# Bedaquiline-Loaded Solid Lipid Nanoparticles Drug Delivery in the Management of Non-Small-Cell Lung Cancer (NSCLC)

**DOI:** 10.3390/ph16091309

**Published:** 2023-09-15

**Authors:** Shehla Nasar Mir Najib Ullah, Obaid Afzal, Abdulmalik Saleh Alfawaz Altamimi, Manal A. Alossaimi, Waleed H Almalki, Abdulaziz Alzahrani, Md. Abul Barkat, Tahani M. Almeleebia, Hanan Alshareef, Eman M. Shorog, Gyas Khan, Tanuja Singh, J. K. Singh

**Affiliations:** 1Department of Pharmacognosy, Faculty of Pharmacy, King Khalid University, Abha 62529, Saudi Arabia; 2Department of Pharmaceutical Chemistry, College of Pharmacy, Prince Sattam Bin Abdulaziz University, Alkharj 11942, Saudi Arabia; obaid263@gmail.com (O.A.); as.altamimi@psau.edu.sa (A.S.A.A.); m.alossaimi@psau.edu.sa (M.A.A.); 3Department of Pharmacology and Toxicology, College of Pharmacy, Umm Al-Qura University, Makkah 21955, Saudi Arabia; whmalki@uqu.edu.sa; 4Pharmaceuticals Chemistry Department, Faculty of Clinical Pharmacy, Al-Baha University, Alaqiq 65779-7738, Saudi Arabia; alzahraniaar@bu.edu.sa; 5Department of Pharmaceutics, College of Pharmacy, University of Hafr Al Batin, Hafar Al-Batin 39524, Saudi Arabia; abulbarkat05@gmail.com; 6Department of Clinical Pharmacy, College of Pharmacy, King Khalid University, Abha 61421, Saudi Arabia; talmelby@kku.edu.sa (T.M.A.); eshorog@kku.edu.sa (E.M.S.); 7Pharmacy Practice Department, Faculty of Pharmacy, University of Tabuk, Tabuk 71491, Saudi Arabia; halsharef@ut.edu.sa; 8Department of Pharmacology, College of Pharmacy, Jazan University, Jazan 45142, Saudi Arabia; gkhan@jazanu.edu.sa; 9Department of Botany, Patliputra University, Patna 800020, India; tanujapatnabotany@gmail.com; 10S.S Hospital and Research Institute, Kankarbagh, Patna 800020, India

**Keywords:** bedaquiline, solid lipid nanoparticles, oral drug delivery, pharmacokinetics, biochemical parameters, non-small-cell lung cancer

## Abstract

Non-small-cell lung cancer (NSCLC) mortality and new case rates are both on the rise. Most patients have fewer treatment options accessible due to side effects from drugs and the emergence of drug resistance. Bedaquiline (BQ), a drug licensed by the FDA to treat tuberculosis (TB), has demonstrated highly effective anti-cancer properties in the past. However, it is difficult to transport the biological barriers because of their limited solubility in water. Our study developed a UPLC method whose calibration curves showed linearity in the range of 5 ng/mL to 500 ng/mL. The UPLC method was developed with a retention time of 1.42 and high accuracy and precision. Its LOQ and LOD were observed to be 10 ng/mL and 5 ng/mL, respectively, whereas in the formulation, capmul MCM C10, Poloxamer 188, and PL90G were selected as solid lipids, surfactants, and co-surfactants, respectively, in the development of SLN. To combat NSCLC, we developed solid lipid nanoparticles (SLNs) loaded with BQ, whereas BQ suspension is prepared by the trituration method using acacia powder, hydroxypropyl methylcellulose, polyvinyl acrylic acid, and BQ. The developed and optimized BQ-SLN3 has a particle size of 144 nm and a zeta potential of (−) 16.3 mV. whereas BQ-loaded SLN3 has observed entrapment efficiency (EE) and loading capacity (LC) of 92.05% and 13.33%, respectively. Further, BQ-loaded suspension revealed a particle size of 1180 nm, a PDI of 0.25, and a zeta potential of −0.0668. whereas the EE and LC of BQ-loaded suspension were revealed to be 88.89% and 11.43%, respectively. The BQ-SLN3 exhibited insignificant variation in particle size, homogeneous dispersion, zeta potential, EE, and LC and remained stable over 90 days of storage at 25 °C/60% RH, whereas at 40 °C/75% RH, BQ-SLN3 observed significant variation in the above-mentioned parameters and remained unstable over 90 days of storage. Meanwhile, the BQ suspension at both 25 °C (60% RH) and 40 °C (75% RH) was found to be stable up to 90 days. The optimized BQ-SLN3 and BQ-suspension were in vitro gastrointestinally stable at pH 1.2 and 6.8, respectively. The in vitro drug release of BQ-SLN3 showed 98.19% up to 12 h at pH 7.2 whereas BQ suspensions observed only 40% drug release up to 4 h at pH 7.2 and maximum drug release of >99% within 4 h at pH 4.0. The mathematical modeling of BQ-SLN3 followed first-order release kinetics followed by a non-Fickian diffusion mechanism. After 24 to 72 h, the IC50 value of BQ-SLN3 was 3.46-fold lower than that of the BQ suspension, whereas the blank SLN observed cell viability of 98.01% and an IC50 of 120 g/mL at the end of 72 h. The bioavailability and higher biodistribution of BQ-SLN3 in the lung tumor were also shown to be greater than those of the BQ suspension. The effects of BQ-SLN3 on antioxidant enzymes, including MDA, SOD, CAT, GSH, and GR, in the treated group were significantly improved and reached the level nearest to that of the control group of rats over the cancer group of rats and the BQ suspension-treated group of rats. Moreover, the pharmacodynamic activity resulted in greater tumor volume and tumor weight reduction by BQ-SLN3 over the BQ suspension-treated group. As far as we are aware, this is the first research to look at the potential of SLN as a repurposed oral drug delivery, and the results suggest that BQ-loaded SLN3 is a better approach for NSCLC due to its better action potential.

## 1. Introduction

Almost 20 million new instances of cancer are diagnosed each year, and nearly 10 million people lose their lives to the illness [1]. Lung cancer is the most common cancer-related mortality, accounting for around 18% of all cancer deaths (1.8 million cases) [1]. The uncontrolled proliferation of cells in lung tissue characterizes aggressive, fast-spreading malignancies. Non-small-cell lung cancer (NSCLC) accounts for more than 85% of all occurrences of lung cancer and is the leading cause of cancer death worldwide [2]. Currently available treatments include surgery, chemotherapy, radiation therapy, targeted therapy, and immunotherapy; the most effective combination depends on the kind, stage, and general health of the patient [2]. The most typical method is chemotherapy or the use of cytotoxic medicines. Repurposing existing drugs is becoming more popular due to the slow pace, expensive research expenses, and high failure rates of new drug discovery [3]. Using drugs for uses different from those for which they were first produced is known as “repurposing” [4]. Thus, various efficient drugs for treating both common and uncommon ailments have been discovered by researchers. Researchers have started repurposing antibiotics for use as cancer therapeutics in early-stage clinical studies as a result of the rise in interest in the potential of antibiotics as anti-cancer medications in recent years [4]. The FDA has authorized the use of bedaquiline (BQ) to treat multidrug-resistant pulmonary tuberculosis (TB) [5]. Tumor development can be slowed by using BQ mechanistic actions, which include targeting mitochondrial ATP-synthase and causing mitochondrial malfunction and ATP depletion [5]. According to research by Wu et al., BQ suppresses mitochondrial respiration in the same way that it suppresses bacterial ATP synthase to prevent the establishment of capillary networks in human lung tumor-associated endothelial cells (HLT-EC) [6]. Recent preclinical studies suggest that BQ offers potential as a cancer treatment [6]. In vitro literature claims that BQ has therapeutic potential against NSCLC [6]. In the course of normal cell metabolism, biochemical redox processes produce free radicals, or reactive oxygen species (ROS) [7]. However, chronic lung illnesses cause oxidative stress due to the excess production of ROS [7]. ROS and their reactive derivatives are produced in significant quantities together with the release of inflammatory mediators, which is what causes the damage to the lungs [8]. By using various biological antioxidant defense mechanisms, including enzymatic and non-enzymatic production, free radical damage can be naturally reduced [9]. Superoxide dismutase (SOD), catalase (CAT), glutathione reductase (GR), and malondialdehyde (MDA), a stable product of free radicals that cause oxidative damage to tissues, are examples of important antioxidant enzymes [10]. Glutathione (GSH) is only one of the many natural and manufactured antioxidant molecules found in the non-enzymatic portion [11]. According to earlier research, increased oxidative stress in lung cancer can decrease antioxidant function. Further, insufficient drug concentration commonly demonstrates limited antitumor action and may lead to resistance; anti-cancer medications must penetrate cancer tissues to achieve the concentration required to efficiently eradicate tumours [12]. The most exciting oral drug delivery system is solid lipid nanoparticles (SLNs), which have high drug loading, extended shelf lives, and production-scale viability [13]. It has been crucial in the delivery of various poorly formulated anti-cancer drugs and even those that reverse cancer cell line resistance [13]. Its formulation has been used for several drug delivery methods and received regulatory agency approval [13]. Additionally, SLNs enable the drug’s progressive release over time and are simple to make using inexpensive raw materials and straightforward manufacturing processes [14]. SLNs, which can also improve the stability of drugs and lengthen their half-life and average residence time in vivo, enable a longer and more controlled release of therapeutics [15]. Additionally, the SLN’s lipid layer and a unique tumour microenvironment may impact the SLN, which can allow drugs to be released via a target action in tumour tissue [15,16]. BQ-loaded SLNs might be used in this fashion to effectively carry the drug into the tumour tissue, perhaps lower the dosage, avoid harming healthy organs or tissues, and possibly even reverse MDR in tumours. This research determined the BQ-loaded SLNs would be more effective than using either BQ suspension in treating lung cancer.

## 2. Results

### 2.1. Method Development for BQ Using UPLC-MS/MS

For the developed UPLC-MS/MS method for BQ, the best collision energy values were observed at 40 eV, a cone voltage of 40 V, a cone gas flow of 50 (L/Hr), a capillary voltage of 2.97 KV, and a source temperature of 150 °C. The optimized mobile phase containing ACN (A) and 0.5% *v*/*v* formic acid (ratio 60:40% *v*/*v*) at a flow rate of 0.6 mL. min^−1^ showed suitable peak shapes for BQ. In method development, BQ exhibits a chromatogram after optimization (Figure 1). The retention time of BQ was found to be 1.42 min. Calibration curves of the drug concentration range of 5 and 500 ng mL^−1^. The regression equation was also measured for the drug, and the correlation coefficient for BQ was found to be 0.9987.

### 2.2. Accuracy

Table 1 shows the individual accuracy results for different quality control samples for BQ. With a high degree of method accuracy, the accuracy estimations for the standard BQ identified as lower quality control (LQC) (100 ng mL^−1^), middle quality control (MQC) (250 ng mL^−1^), and higher quality control (HQC) concentrations (450 ng mL^−1^) showed good percentage recovery between 99.50% and 99.98%, while RSD was less than 2% (0.27–1.25).

### 2.3. Precision

As demonstrated in Table 2 and Table 3, the precision (intra-day and inter-day) findings of BQ include quality control samples. The BQ recovery values for precision ranged from 98.5% to 99.8% and 98.70% to 99.91%, respectively. Additionally, it was found that both repeatability and intermediate precision had RSD percentage values that were less than 2% (RSD) is between 0.12–1.31). The approach devised is incredibly reliable based on these findings.

### 2.4. Limits of Quantitation and Limit of Detection 

The limits of quantitation (LOQ) and limit of detection (LOD) of the system developed for BQ quantification were found to be 10 ng mL^−1^ and 5 ng mL^−1^, respectively. This shows that the system has reasonably high sensitivity.

### 2.5. Choosing Excipients Based on Solubility Studies

After 72 h, solubility had stabilized at its maximum level. According to equilibrium solubility studies conducted on BQ, the order of lipid solubility is as follows: Capmul MCM C10 > Compritol ATO 888 > GMS > Stearic Acid. Due to the high solubilizing ability of BQ (120 mg/mL) in Capmul MCM C10, shown in Table 4, because of the presence of long chain fatty acids. To develop the formulation, capmul MCM C10 was used as the solid lipid. Additionally, the BQ is most soluble in Poloxamer 188, followed by Tween 80, Poloxamer 407, Unitop 100, Phospholipid 90G (PL90G), soy lecithin, and phospholipid 90H (PL90H). The highest equilibrium solubility in surfactant, i.e., Poloxamer 188 (140 mg/mL), and cosurfactant, i.e., PL90G (90 mg/mL), is shown in Table 4. Poloxamer 188 and PL90G were chosen as the surfactant and cosurfactant for developing SLN as a result. Non-ionic surfactants are less hazardous than ionic surfactants after oral ingestion [17]. 

### 2.6. Preparation of the SLNs and Suspension 

The BQ-SLNs were made with lipids from the initial solubility test under high shear homogenization (ultra-turrax) (Ika, Staufen, Germany). The solid lipid melts at the required value of 200–450 mg (10 °C above the lipid melting point). Additionally, 50–140 mg of the surfactant (poloxamer 188) and 35–70 mg of the PL90G (as shown in Table 5) were pre-heated together, then mixed with solid lipid together for 10 min at 9000 rpm, then cooled to ice to produce SLN. The section below shows the formulation properties that were characterized, including PS, PDI, zeta potential, EE, and LC. Another part of the formulation is suspension development, prepared by the trituration method; its details are described in Section 4.2.3.

### 2.7. Characterization of the SLNs and Suspension

#### 2.7.1. Particle Size, Polydispersity Index, Zeta Potential and Structural Morphology

The optimized BQ-SLN3 formulation, which had a particle size of 144 nm and a PDI of 0.324, is shown in Figure 2A. This demonstrated the monodisperse nanostructure of the BQ-loaded SLN3 formulation. According to measurements, BQ-SLN3 has a negatively charged zeta potential of −16.3 mV (as shown in Figure 2B). This may be due to the long-chain fatty acids. The BQ-SLN3 nature was confirmed by TEM to be compatible with their nanometric size range of 127–167 nm (Figure 2C). The TEM and DLS values showed a statistically significant variance overall. This is even more likely given that the TEM images were captured when the formulation was solid, while the zeta sizer measured its hydrodynamic diameter while it was liquid [18]. In contrast, the suspension average PS should be >1000 nm or equal to 1000 nm [19]. The BQ suspension revealed a size of 1180 nm and a PDI of 0.25 (as shown in Figure 3A,B). This can be seen in Figure 3C depicting the TEM image, which shows the spherical shape of particles and uniform morphology. 

#### 2.7.2. Entrapment Efficiency and Loading Capacity 

The EE and LC were calculated for the optimized SLN3 under optimal circumstances using the procedure outlined in the analytical Section 4.2.1. The EE and LC were observed at 92.05 ± 3.12% and 13.33 ± 0.71%, respectively, at the optimal concentration of lipids, surfactants, and cosurfactants. Overall, the findings were consistent with those of other researchers who published their findings and supported that finding, indicating that the techniques used in the development of SLNs and other processes were advantageous [18]. 

### 2.8. Stability Studies of BQ-SLN3 and BQ-Suspension 

Long-term and stressed stability test findings for BQ-loaded SLN3 and BQ suspension at 25 °C/60% RH and 40 °C/75% RH are discussed below.

#### 2.8.1. Particle Size, Size Distribution, and Zeta Potential 

After three months of storage at 25 °C/60% RH, the PS of the optimized BQ-loaded SLN3 varied from 144 ± 10.75 nm to 144.53 ± 6.21 nm (shown in Table 6). PS differences were not statistically significant, and particle growth was extremely slow at 25 °C/60% RH. However, the PDI data show that the observed levels were often low and stable. This may be explained by the PS and PDI being homogenous and stable at 25 °C/60% RH for 90 days (as shown in Table 6). Over a period of 90 days of storage at 25 °C/60% RH, the zeta potential data showed insignificant shifts, and therefore it remained stable, whereas storage conditions at 40 °C/75% RH caused significant variations in PS, PDI, and zeta potential of BQ-SLN3 over 90 days (as shown in Table 7) and remained unstable; this may be due to the system kinetic energy increases leading to increased collisions among particles, resulting in aggregation and increased particle size with a heterogeneous dispersion system and increased zeta potential. Further, the enhancement in zeta potential may be due to the neutralization of the charges on the particle surface. Meanwhile, with the BQ-loaded suspension stored at 25 °C/60% RH for the duration of 90 days, the PS varied from 1180 ± 22.25 nm to 1181.23 ± 20.26 nm (shown in Table 8). However, the PDI data showed 0.25 ± 0.04 to 0.312 ± 0.05 and a zeta potential of −0.0668 ± 0.02 to −0.0610 ± 0.04. Therefore, there was nonsignificant variation observed in PS, PDI, and zeta potential values, and the BQ-suspension remained stable over the storage duration of 90 days, whereas at 40 °C/75% RH, the BQ-loaded suspension found nonsignificant variation in the PS, PDI, and zeta potential values (shown in Table 9). So, it was concluded that the BQ-loaded suspension remained stable at 40 °C/75% RH for up to 90 days in storage conditions.

#### 2.8.2. Encapsulation Efficiency and Loading Capacity 

As indicated in Table 6, the EE of BQ-loaded SLN3 at 25 °C/60% RH ranged from 92.05 ± 6.12% to 91.88 ± 5.23%, while the LC was found to be between 13.33 ± 0.71% to 13.05 ± 0.22%. As a result, at 25 °C/60% RH, the amount of BQ-loaded SLN3 varied inconsistently or irrelevantly during the storage period of 90 days. The EE and LC of the formulations therefore showed insignificant fluctuations at 25 °C/60% RH and remained constant for 90 days throughout the stability investigation, whereas the EE and LC of the optimized BQ-loaded SLN3 at 40 °C/75% RH ranged from 92.05 to 81.33%, and the LC was in the range of 13.33 to 8.91%; these are shown in Table 7. Therefore, these results showed a significant decrease in EE and LC, respectively, at storage temperatures of 40 °C/75% RH and remained unacceptable. Moreover, this may concern polymorphic forms of the lipid and drug leakage. As indicated in Table 8, the EE and LC of BQ-loaded suspension at 25 °C/60% RH ranged from 88.89 ± 8.23% to 87.89 ± 8.24% and 11.43 ± 0.65% to 10.66 ± 0.35%, respectively. The EE and LC of the BQ-suspension therefore showed nonsignificant variation at 25 °C/60% RH and remained constant for 90 days throughout the stability investigation, whereas the EE and LC of the BQ-loaded suspension at 40 °C/75% RH ranged from 88.89 ± 8.23 to 87.12 ± 6.22 and 11.43 ± 0.65 to 9.50 ± 0.33, respectively (Table 9). Therefore, these results showed a nonsignificant decrease in EE and LC at storage temperatures of 40 °C/75% RH and remained acceptable. 

### 2.9. In Vitro Gastrointestinal Stability 

Table 10 shows the PS, PDI, zeta potential, EE, and LC of optimized BQ-SLN3 after treatment with GI fluid at SGF at pH 1.2 and SIF at pH 6.8. During stability testing, there was no significant variation observed before and after exposure to the GI fluids (*p* > 0.05). Moreover, the results indicated that the optimized SLN3 formulation was stable in the mentioned pH conditions, whereas Table 11 shows the PS, PDI, zeta potential, EE, and LC of BQ-suspension after treatment with simulated gastric fluid (SGF) at pH 1.2 and simulated intestinal fluid (SIF) at pH 6.8. The data in Table 11 showed that there was no significant variation observed when BQ suspension was treated with the different pH conditions of GI fluids and revealed it to be stable.

### 2.10. In Vitro Drug Release 

Figure 4 displays the in vitro release of BQ-SLN3 at pH 4 (PBS). The varied drug release characteristics were seen at 25% within the first 3 h by BQ-SLN3 and up to a maximum of 44% with almost sustained release up to 12 h. On the other side, BQ-suspension observed 99% drug release at pH 4 up to 3 h. Figure 5 displays the in vitro drug release of BQ-SLN3 at pH 7.2 (PBS), The BQ-loaded SLN3 showed drug release of 60% within the first 3 h and 98.19% up to 12 h, whereas with the BQ-loaded suspension, only 28% of the drug was released within 3 h and maximum drug release was up to 40% in 4 h. Therefore, Figure 4 is important for BQ-loaded SLN3 because of the maximum drug release found at pH 7.2 and because it is best suited at pH 4. Further, it was concluded that BQ is a component of the SLN3 outer layer and that its release is biphasic, with a 0–3-h initial burst release. The fact that the free drug BQ is present in the external phase and on the surface of SLN3 may be the cause of this burst release. The alteration in lipid and surfactant content in the exterior phase also had an impact on the first burst release rate [17,18]. Because there was a larger concentration of drug in the inner core as the lipid concentration increased, the first burst release rate decreased (for the quantity of excipients, see Table 5). A rise in the drug’s solubility in the external phase coincided with an increase in the surfactant concentration, which also caused an increase in the first burst release rate. This bi-phasic release pattern (burst release) can be employed to increase the drug’s penetration, and sustained release can last for an extended length of time [17,18,19]. The mathematical modeling of the release data from BQ-SLN3 followed first-order release kinetics, exhibiting an R value of 0.955. Furthermore, the Korsemeyer–Peppas model applied to the drug release data indicated that the value of the release exponent (n) for the optimized SLN3 was 0.756, thus indicating drug release via an anomalous (non-Fickian) diffusion mechanism. The observed results are in consonance with the previously published literature reports on several reports [19,20].

### 2.11. Cell Culture and Cell Viability Study

The MTT test was used to study the inhibitory effects of BQ-loaded SLN3 on cell line growth at 24, 48, and 72 h. Figure 6A,B shows that blank SLN-treated cells do not reduce cell viability at the 24 to 48 h time course of exposure, while a mild reduction in cell viability (almost 2%) occurred at the end of 72 h (Figure 6C). BQ-SLN3 were evaluated for their potential to inhibit lung cancer (A549) cell lines to learn more about their suppressive effects. The lower the IC50 number, the greater the drug’s capacity to inhibit cancer cell proliferation. The BQ suppressed the proliferation of the lung cancer (A549) cell lines from 24 to 72 h, according to the IC50 data (Figure 7). BQ-SLN3 had the most inhibitory effects on A549 cells at 72 h (cell survival was 6.121%) (as shown in Figure 6C). After 24 to 72 h, the IC50 value of BQ-SLN3 was 3.46-fold lower than that of the BQ suspension (Figure 7), whereas the blank SLN observed cell viability at 98.01% and an IC50 of 120 µg/mL at the end of 72 h (Figure 7). 

### 2.12. Pharmacokinetic Study 

The pharmacokinetic profiles of the BQ-loaded SLN3 were examined and contrasted with those of the BQ-loaded suspension. Figure 8, which depicts the plasma concentration vs. time relationship throughout the course of the experiment’s 24 h, shows the treatment groups for BQ, SLN3, and suspension. The absorption metrics (Cmax and AUC0-t) revealed a significant difference (*p* < 0.05) between the BQ-SLN3 and the BQ suspension. Comparatively to the suspension, the Cmax and AUC0-t of BQ from SLN3 were about 1.96 and 3.90 times greater, respectively (*p* < 0.01). Furthermore, no variations were seen in the SLN with the Tmax value (*p* > 0.05). Additionally, Table 12 shows various pharmacokinetic parameters with their reported values for comparison. Nanosized SLN3 shape and lipophilic features may facilitate nanocarrier uptake via intercellular and paracellular channels, which may account for the significant increases in drug absorption characteristics observed [20,21].

### 2.13. Biodistribution Studies 

Biodistribution experiments tracked the movement of BQ from a single oral dosage (BQ-SLN3) and BQ suspension in two groups of animals to several target organs. The organs were homogenized, and the resulting supernatant was analyzed for specific findings. Compared to the BQ suspension, BQ delivered by SLN3 was far more likely to reach the tumour cells in the lung than the healthy lung tissue or other important organs. Biodistribution profiles of BQ-loaded SLN3 and BQ suspension are shown graphically in Figure 9.

### 2.14. Biochemical Parameters Determination 

The changes in lipid peroxidation in sera were measured using the MDA determination [9,22]. The MDA content in the sera of the cancer group of rats rapidly increased and was most significantly reduced to reach almost the control value (50.55 ± 3.87 nmol/mg) by BQ-SLN3 (54.2 ± 0.32 nmol/mg) over the BQ-suspension (66.01 ± 3.51 nmol/mg) group of rats. The activity of SOD and catalase and their levels in the sera of the control group, cancer groups, and treated rats were assessed during the progression of the disease. In our study, the SOD activities in the lung cancer group of rats were significantly decreased (4.21 ± 0.51*) over the control group of rats (shown in Table 13). The most significant increase was reached closest to the control group (10.12 ± 0.81 unit/mg) of rats by administering BQ-SLN3 (9.5 ± 0.61 unit/mg), whereas, in the case of CAT, a similar pattern of results was observed. The effect of BQ-SLN3 on GSH and GR in the treated group was significantly improved and was closest to the control group of rats over the cancer group of rats and the BQ suspension-treated group of rats.

### 2.15. Pharmacodynamic Evaluation

To further explore the anti-cancer efficacy of BQ-SLN3 in vivo, an animal model was constructed using Wistar rats carrying A549 cells. At the conclusion of the study, there was a significant weight difference between the normal control group and the negative control group, and no Wistar rats perished. Between the treated groups, the BQ-SLN3 observed the most significant reduction in tumor volume and tumor weight over the BQ suspension-treated group of rats (*p* < 0.05) (shown in Figure 10A,B). However, the rats in the negative control group increased their tumor weight much more than the rats in the normal control group. The BQ-SLN3-administered group exhibited the most significant reduced tumor volume at various day intervals over another treated group, as shown in Figure 10C. 

## 3. Discussion 

The FDA has granted BQ, a drug used to treat tuberculosis (TB), a license to treat cancer. However, due to its low solubility in water, it is challenging to deliver and has inappropriate pharmacokinetics. The UPLC-MS/MS method developed for BQ had calibration curves for a drug concentration range of 5 ng mL^−1^ to 500 ng mL^−1^, and the regression correlation coefficient observed was 0.9987 with a retention time of 1.42. The different quality control samples of BQ showed a high degree of accuracy and precision with good percentage recovery, and the RSD was less than 2%. Furthermore, the LOQ and LOD of the system were found to be 10 ng mL^−1^ and 5 ng mL^−1^, respectively. Therefore, this system has a reasonably high sensitivity. After all, the SLNs have been developed using capmul MCM C10, Poloxamer 188, and PL90G as solid lipids, surfactants, and cosurfactants, respectively, where the suspension is prepared by the trituration method [20]. The developed SLN3 was optimized with a PS of 144 ± 2.75 nm, a PDI of 0.324, a zeta potential of −16.3 ± 0.04 mV, 92.05% EE, and a LC of 13.33%, whereas the BQ suspension was found to have a PS of 1180 nm, a PDI of 0.25, a zeta potential of −0.0668, an EE of 88.89%, and a 11.43% LC. The PS, PDI, zeta potential, EE, and LC did not vary appreciably over time, according to the long-term stability assessment of BQ-SLN3, whereas at stressed conditions, BQ-SLN3 remained unstable. BQ suspension revealed stable long-term and stressed conditions up to 90 days. In addition, gastric stability evaluation of BQ-SLN3 and BQ suspension revealed stability at SGF pH 1.2 and SIF pH 6.8. The in vitro drug release of BQ-SLN3 is to its greatest extent 60% and 28% drug release from the BQ suspension within the first 3 h at pH 7.2 (PBS), whereas at pH 4.0 (PBS), only 25.12% drug release from BQ-SLN3 and 99% drug release from BQ suspension occurred in the first 3 h. Therefore, for BQ-SLN3, pH 7.2 (PBS) is selected as the best in vitro drug release medium. In fact, the free drug BQ, which is present in the external lipid layer and on the surface of the SLN3, may be the cause of this burst release. The alteration in lipid and surfactant content in the exterior phase also had an impact on the burst release rate. Because there was a larger concentration of drug in the inner core as the lipid concentration increased, the first burst release rate decreased (for the quantity of excipients, see Table 5). A rise in the drug’s solubility in the external phase coincided with an increase in the surfactant concentration, which also caused an increase in the first burst release rate. This bi-phasic release pattern (burst release) can be employed to increase the drug’s penetration, and sustained release can last for an extended length of time [20,21]. Moreover, the release pattern followed the first order drug release with a non-Fickian diffusion mechanism. Thus, the findings are in consonance with the published literature [18,20]. The MTT test shows that blank SLN-treated cells do not reduce cell viability over a 24 to 48 h time course of exposure, while a mild reduction in cell viability (almost 2%) occurred at the end of 72 h, whereas the IC50 value of BQ-SLN3 was 3.46-fold lower than that of the BQ suspension. Apart from this, the BQ-loaded SLN3 improved the bioavailability of BQ over the BQ suspension. In the biodistribution study, there was a higher accumulation of BQ from SLN3 in the A549 cancer bearing rat over the BQ suspension. Although the pathophysiology of cancer is still unclear, there is strong evidence that free radicals, especially oxygen radicals, play a significant role in the intricate process of multi-step carcinogenesis. Reversible or irreversible tissue injury is the outcome of oxidative stress, which can be caused by excess production or a lack of antioxidant defense or repair mechanisms (enzymatic or non-enzymatic) [7,8,9,10]. Through disturbance of cellular processes and integrity, this may increase lipid peroxidation and oxidative damage to macromolecules such as lipid, DNA, RNA, and antioxidant enzymes in succeeding cells [9]. In our work, we suggested that illness stages are characterized by excessive, non-specific generation of free radicals. According to the results, lipid peroxidation is still occurring in the affected tissues and is followed by the release of lipid peroxidation products into the bloodstream. In our work, lung cancer stages are characterized by excessive, non-specific generation of free radicals via lipid peroxidation within the influenced tissues. MDA is the key final product of lipid peroxidation; this aldehyde can increase oxidative stress by enhancing the cellular consumption of GSH which is the first antioxidant defense enzyme that converts superoxide radicals to H_2_O_2_ and shields cells from the harmful effects of free radicals [7,8,9,10,11,22]. In diseased situations, there has been evidence of decreased SOD activity. In our experiment, the activity of the antioxidant enzyme SOD was significantly lowered and most significantly reversed by administration of BQ-SLN3, as shown in Table 13. It has been suggested that CAT is crucial to cells’ defense through the detoxification of elevated H_2_O_2_ concentrations. In our present study, disease suggests a decrease in CAT activity due to excess H_2_O_2_ production, and its activity was reversed most effectively by using BQ-SLN3. GSH is a key molecule in redox body homeostasis. GSH can function as a free radical scavenger directly by neutralizing HO• or indirectly by repairing the original harm that HO• caused macromolecules. Its decrease in concentration in the circulation in various disease conditions has been reported in the literature [7]. In our present study, the GSH and GR levels decreased significantly in the lung cancer group of rats over the control rats. According to Sarkar et al., increased production of ROS at a rate that exceeds the capacity to regenerate GSH may be the cause of the decreased GSH level [8,22]. This increased level of lipid oxidation products may be related to the lower availability of NADPH required for the activity of GR to transform oxidized glutathione to GSH. Therefore, administration of BQ-SLN3 most effectively altered its activity to attain the near value of the control group of rats over other-treated groups. When compared to controls, it was found that the lung cancer group of rats had changed MDA, GSH, GR, CAT, and SOD activities; however, these activities simply reversed and returned to the closest point to the normal control value of rats. The previous published literature supports this finding, whereas based on in vitro assessment, it showed improved pharmacodynamic findings, including numerous BQ-loaded SLN3. The BQ-SLN3-administrated group exhibited the most significant reduction in tumor volume at various day intervals over another treated group. However, the rats in the negative control group increased much more tumor weight than the rats in the normal control group, which just reversed in the BQ-SLN3 administration group. Therefore, the in vivo result was in accordance with the cytotoxicity findings on the A549 cell line. Lung cancer treatment with BQ-SLN3 has been demonstrated to be an effective prospective antineoplastic drug. Finally, it can be said that the BQ-SLN3 system’s overall effectiveness against lung cancer was made possible by its nanosized formulation, good stability, increased cytotoxicity, enhanced pharmacokinetics, and tumor volume decrease with adjusted body weight. 

## 4. Materials and Methods

### 4.1. Materials

Bedaquiline (BQ) was purchased from Med Chem Express (Princeton, NJ, USA). Fisher Scientific (Hampton, NH, USA) was used to get HPLC-grade formic acid, acetonitrile (ACN), and water. Capmul MCM C10, compritol ATO 888, glycerol monostearate, and stearic acid were purchased from M/s Gattefosse, Cedex, France. The poloxamers (Kolliphor^®^ P 188 and Kolliphor^®^ P 407) used were purchased from BASF (Mumbai, India). Soy lecithin, phospholipid 90G (PL90G), and phospholipid 90H were purchased from BASF in Mumbai, India. Unitop100, Tween 80, was purchased by Fischer Scientific Pvt. Ltd., India. Local suppliers were sourced for HPLC-grade solvents. All reagents were of analytical quality and obtained from an approved vendor.

### 4.2. Methods 

#### 4.2.1. Method Development for BQ Using Ultra-Performance Liquid Chromatography Tandem Mass Spectrometry (UPLC-MS/MS)

UPLC-MS/MS, a quantitative analytical method [23,24] for BQ, was developed. An analytical C18 column with a particle size of 5 µm (Zorbax^®^, Agilent Corporation, Santa Clara, CA, USA) was used for the chromatographic separation. Acetonitrile (ACN) and 0.5% *v*/*v* formic acid (A) 60: 40 (B) *v*/*v* were combined as the solvent mixture’s mobile phase, moving at a rate of 0.6 mL/min. UV detection was performed at 333 nm, and the BQ retention time was 1.42 min, all at a working temperature of 25 °C. By precisely diluting the stock solution (1 µg/mL) of formic acid (HPLC grade) and ACN, the BQ calibration curve was made to have the following concentrations: BQ at 1, 5, 10, 20, 50, 100, 250, and 500 ng/mL. While Quant LynxVR (Waters) was used for quantification by comparing the results of the ratios of BQ peak areas to a previously obtained calibration curve under the same experimental conditions, electrospray ionization was employed for detection in a positive ion multiple reaction monitoring mode. The new method’s linearity range was examined for accuracy, precision, limit of quantification (LOQ), and limit of detection (LOD). Furthermore, using Equations (1)–(3), accuracy, LOD, and LOQ are determined.
Accuracy = [(measured concentration − nominal concentration)/nominal concentration] × 100(1)
LOD = Std. Deviation × 3.3/Slope(2)
LOQ = Std. Deviation × 10/Slope(3)

#### 4.2.2. Selection of Ingredients

##### Solubility Study

The equilibrium solubility of BQ in several solid lipids, surfactants, and cosurfactants. Mono-diglycerides of medium-chain fatty acids (Cap MCM-C10), stearic acid, GMS, and Compritol ATO 888 (glycerol dibehenate ep/glyceryl dibehenate) were used as the solid lipids. Tween 80 (T-80), unitop 100, poloxamer, soy lecithin, phospholipid 90 H (PL90H), and phospholipid 90 G (PL90G) were investigated as surfactants and cosurfactants. To sum up, culture tubes containing surfactants and cosurfactants were maintained at a constant temperature and continuously stirred at 25 ± 1 °C for 72 h in a water bath, while solid lipid excipients were heated to a temperature of around 10 °C over their melting point [6,7,8]. Additional BQ was added to all the solid lipids, surfactants, cosurfactants, and phospholipids, and the mixture was stirred consistently at 25 ± 1 °C for 72 h [25]. These samples were also centrifuged at 2000 rpm for 15 min. Filtration of the supernatant through a membrane with a pore size of 0.45 µm was followed by dilution of an aliquot of the resulting filtrate, which was quantified with UPLC-MS/MS analysis.

#### 4.2.3. Preparation of the SLNs and Suspension 

To prepare SLNs, hot microemulsification and solvent diffusion are used [26]. This approach uses lipids and the functional ranges that were chosen from the preliminary solubility investigation of the drug [15]. Cap MCMC10 (i.e., 200 to 450 mg) and PL90G (i.e., 35 mg to 70 mg) were chosen as the solid lipid and co-surfactant in this, and both were heated to 80 °C. Next, the predetermined amount of drug (i.e., 5 mg) was added, and gentle mixing was continued until the drug was completely dissolved in lipid. Additionally, 15 mL of distilled water was used to create a 5 to 6% *w*/*v* (i.e., 50 to 140 mg) aqueous solution of poloxamer 188 at 80 °C. The aqueous phase was then added to the organic phase while being continuously homogenized at a speed of 9000 rpm for 4 min to obtain a uniform dispersion. SLNs were prepared by performing solvent diffusion with an excess of water (15 mL), then agitating the mixture at 2000 rpm in an ice bath for 1–5 h. To eliminate the unencapsulated drug, the formulation was centrifuged for 30 min at 22,000× *g* (at 4 °C), and the supernatant was taken for further study. Another part of the formulation is suspension development, prepared by the trituration method [19]. In order to create a smooth paste, 0.12 g of acacia powder, 0.25 g of hydroxypropyl methylcellulose (HPMC), 0.125 g of polyvinyl acrylic acid (PVA), and 5 mg of BQ were triturated in a mortar and pestle. Then, 0.25 g of methyl paraben was progressively added while being constantly stirred, and the mixture was combined with a 0.25% solution of tween 80. An appropriate amount of peppermint oil and purified water were then added. The mixture was poured into a 25 mL container, filled to volume with distilled water, and rapidly shaken for two minutes.

#### 4.2.4. Characterization of the SLNs and Suspension

##### Particle Size, Polydispersity Index, Zeta Potential, and Structural Morphology 

Using the dynamic light scattering method (Zeta sizer Nano ZS, Malvern Instruments Ltd., Malvern, UK), the zeta potential, the polydispersity index (PDI), which is a measure of how uniform the sample is, and the particle size (PS) were all measured [21,26]. A quantity of 1500 µL of deionized water was used to dilute an aliquot of BQ-loaded SLNs and BQ-loaded suspension (20 µL) at 25 °C (n = 3). The morphology of the developed BQ-SLNs and BQ-loaded suspension was observed using a transmission electron microscope (TEM) (JEM-2100F, M/s Jeol, Tokyo, Japan). The SLN dispersion was applied on copper grids and diluted 100 times with distilled water before being stained with a 1% phosphotungstic acid solution and subsequently studied using an electron microscope (JEM-2100F, M/s Jeol, Tokyo, Japan).

##### Encapsulation Efficiency (EE) and Drug Loading Capacity (LC)

The quantity of BQ added to the SLNs and suspension was measured using the direct vesicle lysis method [21,26]. In a nutshell, 2 mL ACN and 980 µL of HPLC water were added individually to 20 µL of the SLN formulation and suspension to lyse it. To eliminate any remaining undissolved material, the sample was centrifuged for 30 min at 22,000× *g* (at 4 °C), and the supernatant was then analyzed using the previously mentioned UPLC-MS/MS method (Section 4.2.1). According to Equations (4) and (5), the following values for EE% and LC were determined:(4)$$\mathrm{EE}\%=\frac{\mathrm{BQ}\;\mathrm{encapsulated}\;\mathrm{in}\;\mathrm{SLN}\;\mathrm{or}\;\mathrm{suspension}\;}{\mathrm{Initial}\;\mathrm{drug}\;\mathrm{added}}\times 100$$
(5)$$\mathrm{LC}\;\left(\%\right)=\frac{\mathrm{BQ}\;\mathrm{encapsulated}\;\mathrm{in}\;\mathrm{SLN}\;\mathrm{or}\;\mathrm{suspension}}{\mathrm{Total}\;\mathrm{amount}\;\mathrm{of}\;\mathrm{SLN}\;\mathrm{weight}\;\mathrm{or}\;\mathrm{Suspension}\;}\times 100$$

#### 4.2.5. Stability Studies of SLNs and Suspension

To evaluate the stability of the BQ-loaded SLNs and suspension, samples were stored for a total of 90 days at 25 °C/60% RH and 40 °C/75% RH. The PS, PDI, zeta potential, EE, and LC were measured for samples collected after 0, 15, 30, 60, and 90 days using the methods outlined above. The drug content of the BQ-loaded SLNs and BQ suspension was determined using the lysis technique [14,19] and the UPLC-MS/MS method (described in Section 4.2.1). In triplicate, each experiment was carried out. 

#### 4.2.6. In Vitro Gastrointestinal Stability

The in vitro gastrointestinal stability of BQ-loaded SLNs and suspension was assessed by exposing the prepared SLN formulation and BQ suspension (aliquot 5 mL) to 250 mL of simulated gastric and intestinal fluids for 2 and 6 h, respectively [20,21]. The PS, PDI, zeta potential, EE, and LC of the sample were determined at the specified time points.

#### 4.2.7. In Vitro Drug Release 

The in vitro drug release profile of BQ-loaded SLNs and BQ suspension was studied using a dialysis cassette (Slide-ALyzerTM G2, Thermo-Scientific, Waltham, MA, USA) with a molecular weight cut-off of 7000 Da [21]. The dialysis cassette was filled with 1 mL of BQ-loaded SLNs and BQ-loaded suspensions individually placed in 95 mL of phosphate buffered saline (PBS) of pH 4 and 7.2 at 37 °C while being constantly stirred. The releasing medium also contained 0.1% poloxamer-188 to aid in the solubilization of BQ. At regular intervals, 2 mL release medium was collected, and fresh medium was added to maintain the volume of release medium. The concentration of BQ in the release samples was evaluated using the previously published UPLC-MS/MS method (Section 4.2.1). For the data, the cumulative amount of drug released was calculated and plotted versus time.

#### 4.2.8. Studies of Cell Viability and Cell Culture 

##### Cytotoxicity Studies 

The MTT assay was used to test the cytotoxicity of BQ-loaded SLNs and BQ suspension in a test tube. In 96-well tissue culture plates, 100 µL of A549 cells were seeded at a density of 1 × 10^5^ cells/mL and incubated for the whole night at 37 °C with 5% CO_2_. Then, treatment doses of freshly generated BQ-loaded SLNs and BQ suspension (0–60 µM in 100 µL) were added separately, with 50% dimethyl sulfoxide (DMSO) serving as a positive control. After a 72 h treatment period, the treatments were withdrawn from the experiment, and 100 µL of 1 mg/mL MTT solution was added. To dissolve the formazan crystal, the MTT solution was aspirated after 2 h of incubation at 37 °C and replaced with 100 µL of DMSO. The absorbance of the solubilized dye, which was determined at 570 nm using a plate reader (Synergy H1, BioTek, Winooski, VT, USA), is associated with the quantity of living cells in each well. The proportion of living cells was estimated using the absorbance ratio between treatment groups and control wells that were not treated. The half-inhibitory concentration (IC_50_) was calculated using non-linear regression analysis in GraphPad [27].
(6)$$\%\;cell\;inhibition=100-\left\{\frac{Test}{Control}\right\}\times 100$$

Additionally, the IC_50_ was calculated using the Graph Pad Prism.

#### 4.2.9. Animal Studies 

##### Pharmacokinetic Study 

For the suggested experimental investigation, a male Wistar rat weighing 110–125 g was chosen. It acclimated to a temperature of 25 ± 2 °C and a 12-h light–dark cycle, and was given the normal rat food and water as needed. The Institutional Animal Ethical Committee regulations, established by the Committee for the Control and Supervision of Experiments permitted through collaboration with T.P.S. College (Patna, India) and SS Hospital and Research Institute (Patna, India), on Animals (protocol number: 1840/PO/ReBi/S/15/CPCSEA), were examined and approved throughout the entirety of the experimental procedures. Before the experiment, the male Wistar rats underwent fasting and had unrestricted access to water for more than 24 h. The Wistar rat was divided into two groups of six rats, each at random. BQ suspension was given to the first group (5 mg/kg), and BQ-SLNs were given to the second group (5 mg/kg). Through the oral route, all formulations were administered. A heparin-coated syringe was used to draw 1 mL of blood from the marginal ear vein 15, 30, 1, 2, 3, 6, and 24 h following the injection. The supernatant plasma was separated from the blood samples by centrifuging them at 5000 rpm for 10 min. To 25 mL of acetate buffer (pH 3.6), 475 µL of plasma were added. The mixture was first vortexed for 30 s, then 3 mL of the extract solvent (ACN % *v*/*v*) was added. Afterward, the mixture was vortexed for an additional 3 min, and then it was centrifuged for 4 min at 7000 rpm. The organic solvent was then evaporated using a nitrogen flow after 2 mL of the supernatant had been removed. After being dissolved in 100 µL of mobile phase, the dry residue was centrifuged at 8000 rpm for 10 min. The previously mentioned UPLC-MS/MS analytical technique was used to determine the amount of BQ present in the plasma sample.

##### Biodistribution Studies 

The six male Wistar rats were separated into two groups. In the first and second groups, respectively, a single dosage of BQ suspension and BQ-loaded SLN3 was administered. Organs such as the liver, spleen, kidney, lung tumour, heart, lungs, and brain were immediately taken out of each group (n = 3) and dried using tissue paper, weighed properly, and then homogenized with 1 mL of ice-cold KCl solution per 0.5 g of tissue. The supernatant was examined after being kept at −40 °C for a period [21]. The analytical process outlined in Section 4.2.1 was used in the tests.

##### Biochemical Measurements 

Blood was taken from the tail vein of the rats after diethyl-ether anesthesia using capillary tubes heparinized and stored at 4 °C for further use. Moreover, the serum was separated by centrifugation (5000 rpm) at 37 °C and various antioxidant parameters. MDA, SOD, CAT, GSH, and GR activity are measured by the amount of NADPH [7,8,9,10,11].

##### Pharmacodynamic Evaluations

The right forelimb of a Wistar rat received a subcutaneous injection of about 2.5 million A549 tumour cells. Using the formula V = 0.5 × a × b^2^, the tumor volume (V) was calculated, where “a” is entitled to the maximum perpendicular diameter and “b” denotes the minimum perpendicular diameter, using dial calipers. When the tumor volume reached 100 ± 20 mm^3^, rat models with tumors were used for the next experiments. The Wistar rats were separated into five groups, each with six animals. Groups I and II were normal control groups, while Group II was a negative control. A daily dosage of 0.9% *w*/*v* normal saline was given to Group III as the negative control group. Group IV of rats is the negative control group receiving a BQ suspension (5 mg/kg) in a single dose of 5 mL/kg/day p.o. for 28 days. Group V is the negative control group of rats getting BQ-SLNs (5 mg/kg) in a single dose of 5 mL/kg/day p.o. for 28 days. All the treated rats were orally administered a 1 mL volume of formulation. The drug was given three times weekly throughout the course of a 28-day treatment cycle. Tumor-bearing rats were monitored daily for their nutrition, mental health, and everyday activities. The body weight and tumor volume of the Wistar rats were observed as a conclusion to the investigation. The rats that had tumors were euthanized 48 h after the final treatment, the tumors were entirely excised, and the tumors were weighed using an electronic balance. 

##### Data Analysis

All data are shown as means with a standard deviation (SD) or standard error (SE) of the mean, unless otherwise specified in the Standard Error of the Mean (SEM). GraphPad Prism (Version 9.0, GraphPad Inc., San Diego, CA, USA) was used to do an unpaired Student t-test, a one- or two-way analysis of variance (ANOVA), and a Tukey’s multiple comparisons test to see if there was statistical significance.

## 5. Conclusions

The UPLC-MS/MS method for BQ observed a retention time of 1.42. Calibration curves of the drug concentrations ranged between 5 and 500 ng/mL with a correlation coefficient of 0.9987. Further, the system has reasonably high sensitivity, accuracy, and precision. The LOQ and LOD were found to be 10 ng mL^−1^ and 5 ng mL^−1^, respectively. Based on the solubility study results, capmul MCM C10, Poloxamer 188, and PL90G were selected as solid lipids, surfactants, and cosurfactants, respectively, for the SLN development. The optimized BQ-SLN3 formulation, which has a particle size of 144 nm and a PDI of 0.324, has a negatively charged zeta potential of −16.3 mV. The BQ SLN3 nature was confirmed by TEM to be compatible with their nanometric size. The optimized SLN3 exhibited long-term stability at 25 °C/60% RH and was in vitro gastrointestinally stable at pH 1.2 (SGF) and 6.8 (SIF), respectively, whereas BQ suspensions were observed to be stable at 25 °C/60% RH and 40 °C ± 2 °C/75% RH. The gastric stability evaluation of BQ suspension revealed it to be stable at SGF pH 1.2 and SIF pH 6.8. The in vitro drug release characteristics of BQ-SLN3 are to a greater extent 60% within the first 3 h and 98.19% up to 12 h at pH 7.2 with the first-order drug release with a non-Fickian diffusion mechanism, whereas BQ suspensions were found to have 99.05% drug release within the first 3 h and a maximum of 99.87% drug release up to 4 h at a pH of 4.0. After 24 to 72 h, the IC50 value of BQ-SLN3 was 3.46-fold lower than that of the BQ suspension, whereas the blank SLN observed cell viability of 98.01% and an IC50 of 120 µg/mL at the end of 72 h. BQ-SLN3 can both boost a drug’s effectiveness and decrease its toxicity. The lung cancer group of rats had changed MDA, GSH, GR, CAT, and SOD activities; however, these activities simply reversed and returned to the closest point of the normal control group of rats by administering BQ-SLN3. Finally, the in vivo experiment confirms the in vitro finding that BQ-SLN3 exhibited an appropriate pharmacokinetic behaviour with the greatest tumour-volume reduction and higher BQ accumulation into the lung tumour. As per the findings of this study, an effective loading of BQ into SLN3 is a promising novel approach for anti-cancer therapeutics, which will ultimately lead to a more potent and less harmful anticancer treatment.

## Figures and Tables

**Figure 1 pharmaceuticals-16-01309-f001:**
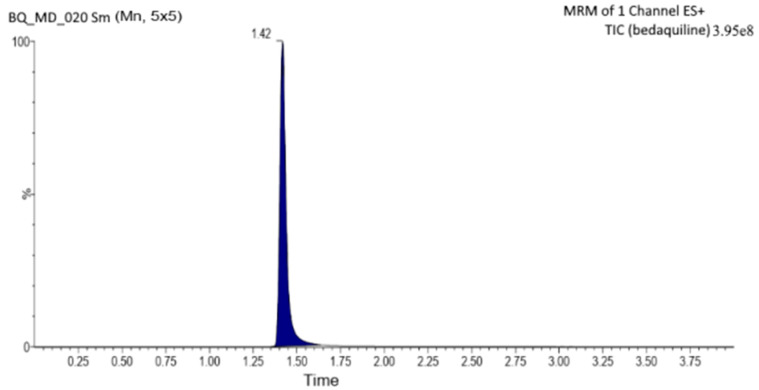
It represents the chromatograms obtained after multiple reaction monitoring (MRM) of bedaquiline (BQ).

**Figure 2 pharmaceuticals-16-01309-f002:**
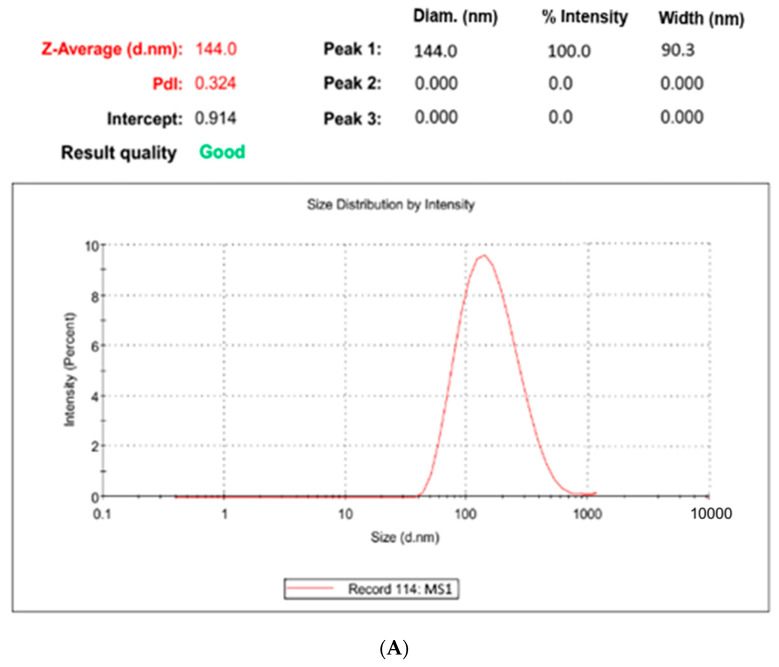
(**A**) It represents the optimized BQ-loaded SLN3 particle size distribution (BQ: bedaquiline; SLN3: optimized solid lipid nanoparticles). (**B**) It represents the zeta potential of optimized BQ-loaded SLN3 (BQ; bedaquiline; SLN3; optimized solid lipid nanoparticles). (**C**) It represents the transmission electron microscopy (TEM) of optimized BQ-SLN3.

**Figure 3 pharmaceuticals-16-01309-f003:**
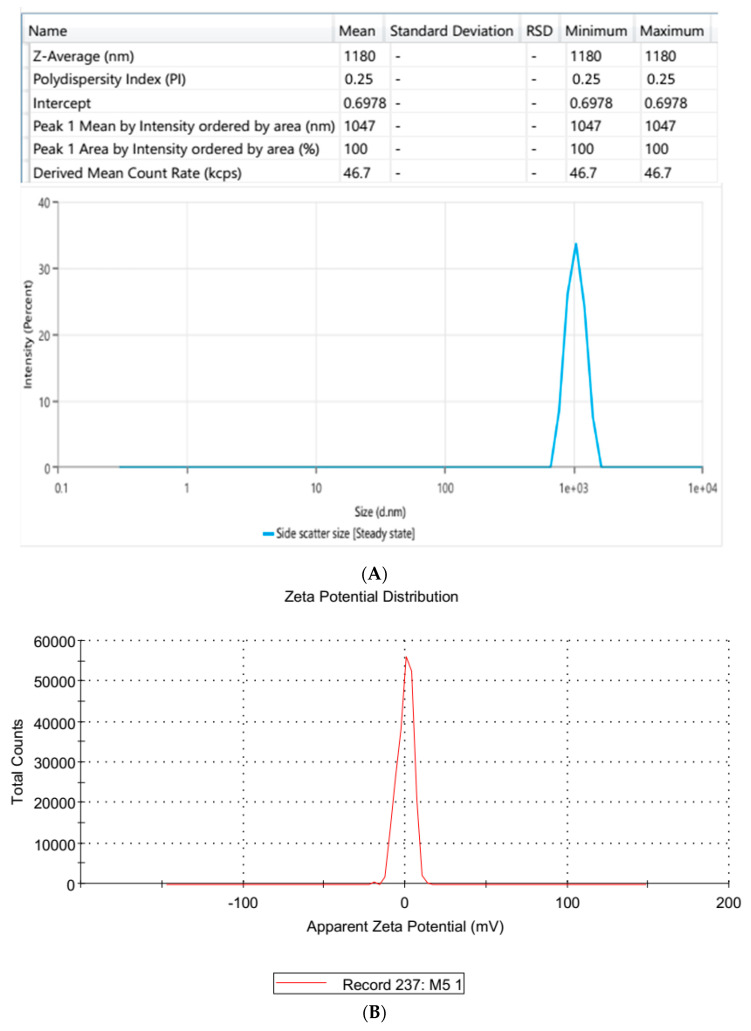
(**A**) It represents the BQ-loaded suspension particle size distribution (BQ: bedaquiline). (**B**) It represents the zeta potential of BQ-loaded suspension (BQ: Bedaquiline). (**C**) It presents the transmission electron microscopy of a BQ-loaded suspension.

**Figure 4 pharmaceuticals-16-01309-f004:**
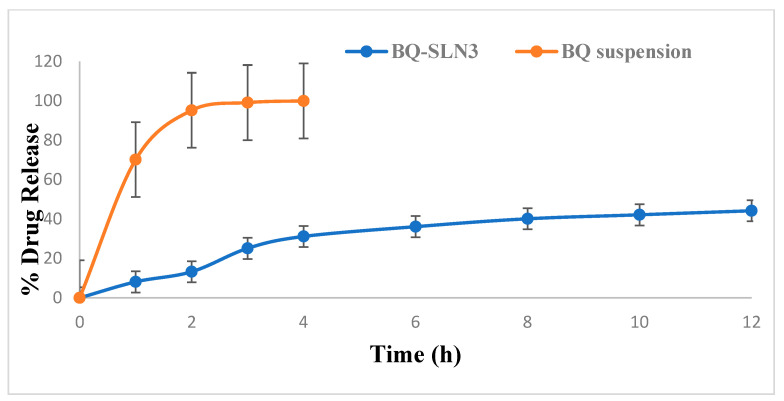
In vitro drug release profile of BQ from suspension and SLN3 at pH 4.0 (data expressed as mean ± S.D (n = 6) (BQ: bedaquiline; SLN3: optimized solid lipid nanoparticles).

**Figure 5 pharmaceuticals-16-01309-f005:**
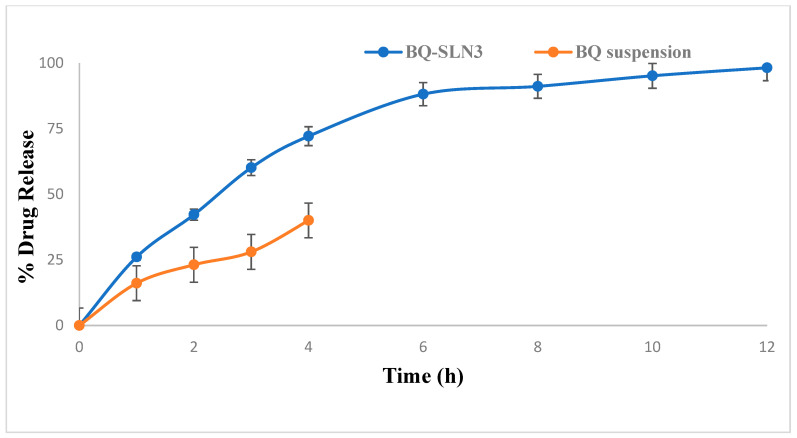
In vitro drug release profile of BQ from suspension and SLN3 at pH 7.2 (data expressed as mean ± S.D (n = 6) (BQ: bedaquiline; SLN3: optimized solid lipid nanoparticles).

**Figure 6 pharmaceuticals-16-01309-f006:**
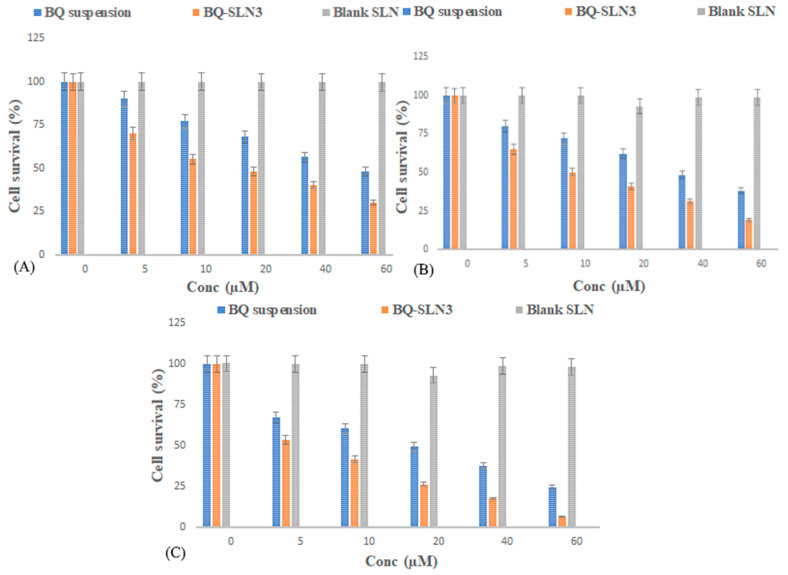
(**A**–**C**) It represents the assessment of cell survival of BQ suspension, BQ-SLN3, and blank SLN at the concentration range 0 to 60 µM on A549 cells (**A**) for 24 h, (**B**) for 48 h, and (**C**) for 72 h. All the values are presented as the mean ± SEM of three independent experiments. A one-way ANOVA was used for the statistical analysis.

**Figure 7 pharmaceuticals-16-01309-f007:**
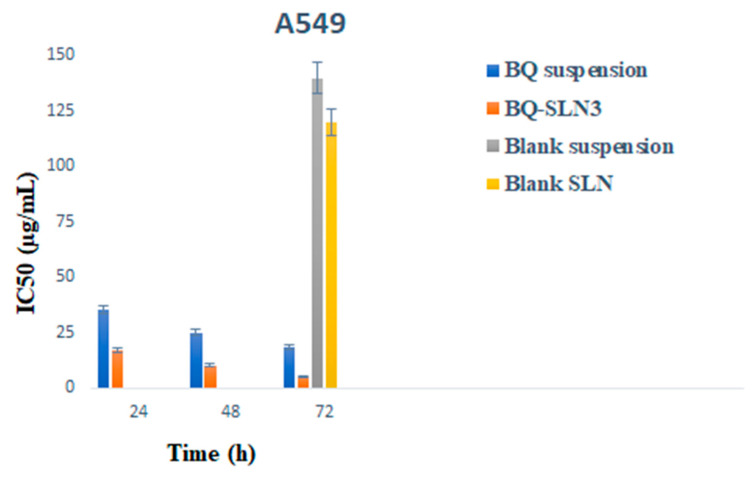
It represents the IC50 of BQ-loaded suspension, BQ-loaded SLN3, blank suspension, and blank SLN. BQ-loaded suspension and BQ-loaded SLN3 show significant (*p* < 0.05) cytotoxicity compared to blank suspension and blank SLN after 24–72 h. Data are expressed as mean ± SD (n = 6). Blank suspension and Blank SLN observed toxicity in 72 h only at concentrations of 140 µg/mL and 120 µg/mL respectively (BQ: bedaquiline; SLN3: optimized solid lipid nanoparticles). IC_50_ = half maximal inhibitory concentration.

**Figure 8 pharmaceuticals-16-01309-f008:**
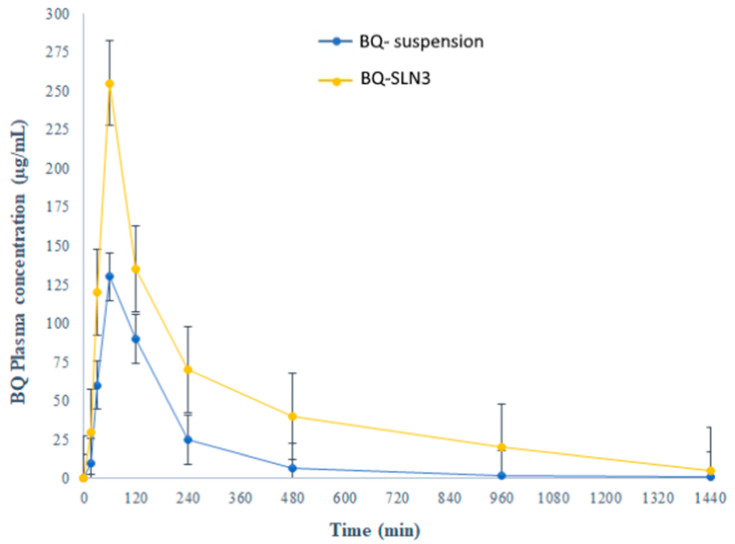
In vivo pharmacokinetic profile of BQ-SLN3 versus BQ-loaded suspension (BQ: bedaquiline; SLN3: optimized solid lipid nanoparticles; all the values were presented as the mean ± SEM of 6 independent experiments).

**Figure 9 pharmaceuticals-16-01309-f009:**
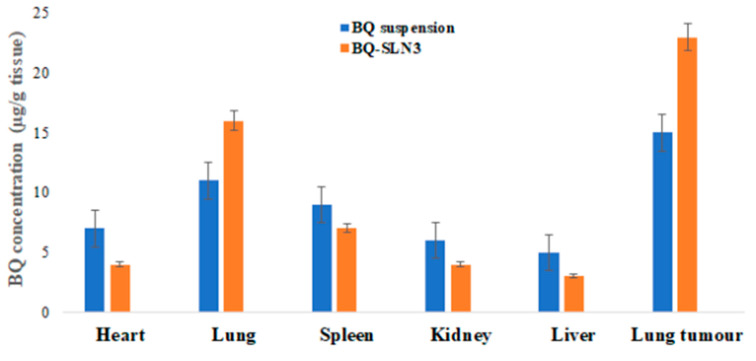
Biodistributions of BQ suspension and BQ-loaded SLN3; enhanced uptake of BQ-SLN by lung tumour followed by lungs (BQ: bedaquiline; SLN3: optimized solid lipid nanoparticles; All the values were presented as the mean ± SEM of 6 independent experiments).

**Figure 10 pharmaceuticals-16-01309-f010:**
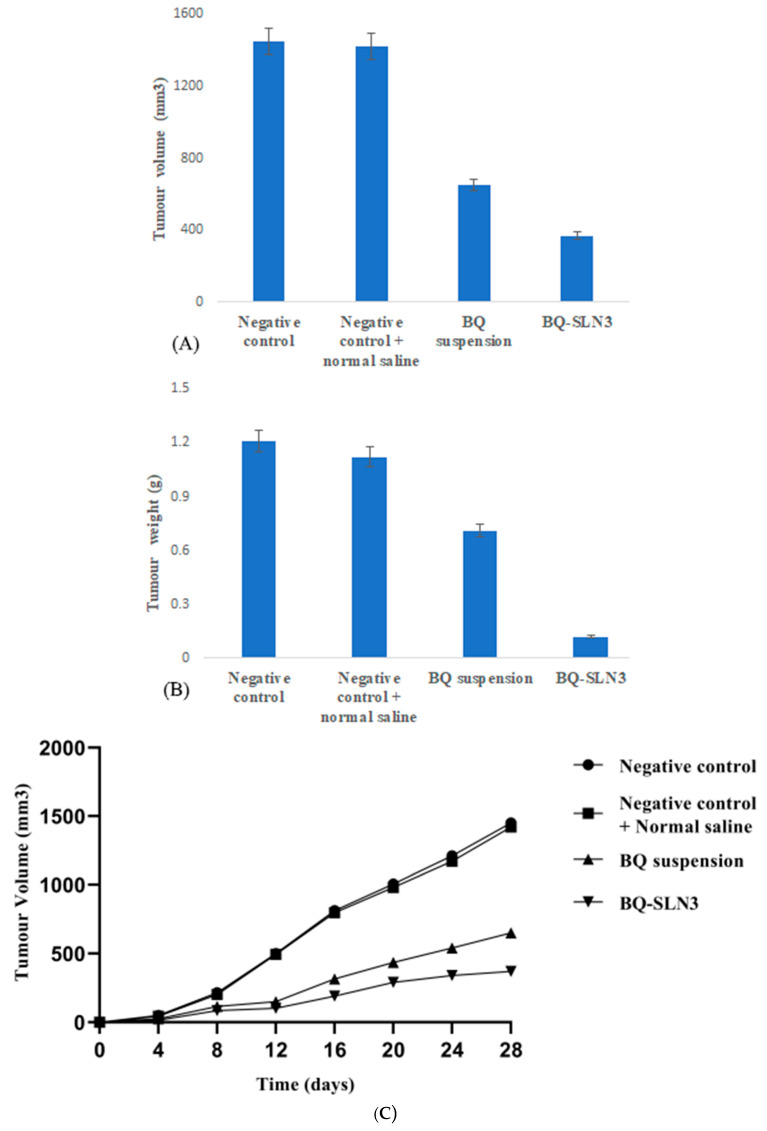
(**A**,**B**) Antitumor efficacy of negative control with saline, BQ suspension, and BQ-loaded SLN3 in an in vivo Wistar rat A549 tumor model (**A**) Tumor volume of an A549-bearing Wistar rat; (**B**) Changes in tumour weight at the end of 28 days (BQ: bedaquiline; SLN3: optimized solid lipid nanoparticles; all the values were presented as the mean ± SEM of 6 independent experiments). (**C**) Tumor volume reduction and development of a negative control with saline, BQ suspension, and BQ-loaded SLN3 on an in vivo Wistar rat A549 tumor model with day’s intervals (BQ: bedaquiline; SLN3: optimized solid lipid nanoparticles; all the values were presented as the mean ± SEM of 6 independent experiments).

**Table 1 pharmaceuticals-16-01309-t001:** Accuracy data of the developed UPLC-MS/MS method for BQ.

Standard Conc.	Levels (%)	Predicted Conc. (ng mL^−1^)	Recovered Conc. (ng. ml^−1^)	Recovery (%)	RSD (%)
BQ (500 ng mL^−1^)	LQC: 50	100	101.0 ± 1.03	99.50	0.27
	MQC: 100	250	251.2 ± 1.37	99.82	0.26
	HQC: 150	450	452.1 ± 1.08	99.98	1.25

LQC: Lower quality control; MQC: Middle quality control; HQC: Higher quality control; BQ: Bedaquiline.

**Table 2 pharmaceuticals-16-01309-t002:** Intra-day precision data of the developed UPLC-MS/MS method for BQ.

Standard Conc. (ng mL^−1^)	Recovered Conc. (ng mL^−1^) ± S.D.	Recovery (%)	RSD (%)
LQC: 100	99.2 ± 3.51	98.5	0.25
MQC: 250	251.5 ± 2.2	99.6	0.32
HQC: 450	451.1 ± 2.51	99.8	0.91

LQC: Lower quality control; MQC: Middle quality control; HQC: Higher quality control; BQ: Bedaquiline.

**Table 3 pharmaceuticals-16-01309-t003:** Inter-day precision data of developed UPLC-MS/MS method for BQ.

Standard Conc. (ng mL^−1^)	Recovered Conc. (ng mL^−1^) ± S.D.	Recovery (%)	RSD (%)
LQC: 100	99.24 ± 3.12	98.70	0.41
MQC: 250	249.31 ± 2.21	98.85	0.12
HQC: 450	450.4 ± 2.07	99.91	1.31

LQC: Lower quality control; MQC: Middle quality control; HQC: Higher quality control; BQ: Bedaquiline.

**Table 4 pharmaceuticals-16-01309-t004:** Solubility of BQ in various excipients.

Solubilizers (Surfactant/Co-Surfactant)	Solubility (mg/mL)
Poloxamer 188	120 ± 4.11
Tween 80	110 ± 5.31
Poloxamer 407	85 ± 2.24
Unitop100	70 ± 4.32
Phospholipid 90 G	90 ± 5. 41
Soy lecithin	76 ± 3.11
Phospholipid 90 H (PL90H),	80 ± 4.24
Solubility of BQ in Lipids	
Capmul MCM C10	130 ± 6.02
Compritol ATO 888	90 ± 5.25
GMS	75 ± 3.41
Stearic Acid	55± 3.21

GMS: Glycerol monostearate; BQ: Bedaquiline.

**Table 5 pharmaceuticals-16-01309-t005:** Quantity of excipient used in solid lipid nanoparticle development.

Formulation Code	Quantity of Solid Lipid (mg)	Quantity of PL90G (mg)	Quantity of Poloxamer 188 (mg)	Particle Size (nm)	PDI
BQ-SLN1	220	35	50	180 ± 12.65	0.421 ±0.05
BQ-SLN2	260	70	140	195 ± 13.76	0.46 ± 0.04
BQ-SLN3	350	55	100	144 ± 10.75	0.324 ± 0.03
BQ-SLN4	300	45	120	175 ± 8.51	0.521 ± 0.06
BQ-SLN5	400	65	75	210 ± 11.31	0.542 ± 0.02

BQ: Bedaquiline; SLN: Solid lipid nanoparticles; PDI: polydispersity index.

**Table 6 pharmaceuticals-16-01309-t006:** Long-term stability data of BQ-SLN3 stored at 25± 2 °C/60%RH (PS, PDI, zeta potential, EE, and LC) (n = 3).

Time (Day)	Size (nm)	PDI	Zeta Potential (mV)	EE (%)	LC (%)
0	144 ± 10.75	0.324 ± 0.03	−16.3 ± 0.04	92.05 ± 6.12	13.33 ± 0.71
15	144.21 ± 8.05	0.326 ±0.04	−16.28 ±0.05	92.03 ± 4.16	13.21 ± 0.61
30	144.35 ± 6.41	0.329 ±0.06	−16.26 ± 0.07	92.0 ± 6.23	13.17 ± 0.43
60	144.42 ± 7.31	0.331 ±0.03	−16.24 ± 0.06	91.98 ± 5.51	13.12 ± 0.32
90	144.53 ± 6.21	0.352 ± 0.04	−16.01 ± 0.04	91.88 ± 5.23	13.05 ± 0.22

EE: Entrapment efficiency; LC: Loading capacity; PDI: polydispersity Index; BQ: Bedaquiline; SLN: Solid lipid nanoparticles.

**Table 7 pharmaceuticals-16-01309-t007:** Particle size, size distribution, zeta potential, entrapment efficiency, and loading capacity of BQ-SLN3 following storage at 40 °C ± 2 °C/75% RH (n = 3).

Time (Day)	Size (nm)	PDI	Zeta Potential (mV)	EE (%)	LC (%)
0	144 ± 10.75	0.324 ± 0.03	−16.3 ± 0.04	92.05 ± 6.12	13.33 ± 0.71
15	146.41 ± 9.05	0.45 ± 0.04	−14.12 ±0.05	90.03 ± 3.16	11.32 ± 0.52
30	149.21 ± 8.23	0.56 ± 0.07	−12.34 ± 0.09	87.05 ± 5.11	10.22 ± 0.32
60	152.42 ± 10.31	0.62 ± 0.09	−9.14 ± 0.06	84.98 ± 4.32	9.33 ± 0.31
90	158.23 ± 11.21	0.71 ± 0.10	−7.05 ± 0.03	81.33 ± 4.11	8.91 ± 0.11

EE: Entrapment efficiency; LC: Loading capacity; PDI: polydispersity Index; BQ: Bedaquiline; SLNs: Solid lipid nanoparticles.

**Table 8 pharmaceuticals-16-01309-t008:** Long-term stability data of BQ-loaded suspension stored at 25 ± 2 °C/60%RH (PS, PDI, zeta potential, EE, and LC) (n = 3).

Time (Day)	Size (nm)	PDI	Zeta Potential (mV)	EE (%)	LC (%)
0	1180 ± 22.25	0.25 ± 0.04	−0.0668 ± 0.02	88.89 ± 8.23	11.43 ± 0.65
15	1180.32 ± 20.12	0.27 ± 0.07	−0.0660 ± 0.03	88.63 ± 7.23	11.23 ± 0.42
30	1180.75 ± 19.35	0.291 ± 0.08	−0.0650 ± 0.04	88.43 ± 8.33	11.10 ± 0.32
60	1180.96 ± 21.35	0.299 ± 0.05	−0.0630 ± 0.03	88.23 ± 7.33	10.97 ± 0.22
90	1181.23 ± 20.26	0.312 ± 0.05	−0.0610 ± 0.04	87.89 ± 8.24	10.66 ± 0.35

EE: Entrapment efficiency; LC: Loading capacity; PDI: polydispersity Index; BQ: Bedaquiline.

**Table 9 pharmaceuticals-16-01309-t009:** Particle size, size distribution, zeta potential, entrapment efficiency, and loading capacity of BQ-loaded suspension following storage at 40 °C ± 2 °C/75% RH (n = 3).

Time (Day)	Size (nm)	PDI	Zeta Potential (mV)	EE (%)	LC (%)
0	1180 ± 22.25	0.25 ± 0.04	−0.0668 ± 0.02	88.89 ± 8.23	11.43 ± 0.65
15	1180.8 ± 18.23	0.28 ± 0.03	−0.0662 ± 0.03	88.65 ± 7.34	11.22 ± 0.45
30	1181.21 ± 19.33	0.32 ± 0.09	−0.0655 ± 0.06	88.12 ± 6.22	10.43 ± 0.41
60	1182.22 ± 20.23	0.36 ± 0.07	−0.0646 ± 0.07	87.67 ± 7.34	9.92 ± 0.23
90	1182.76 ± 19.45	0.41 ± 0.06	−0.0640 ± 0.05	87.12 ± 6.22	9.50 ± 0.33

EE: Entrapment efficiency; LC: Loading capacity; PDI: polydispersity Index; BQ: Bedaquiline.

**Table 10 pharmaceuticals-16-01309-t010:** In vitro gastrointestinal stability of optimized BQ-SLN3 in various gastric media.

Stability Parameters	SGF (pH 1.2)		SIF (pH 6.8)	
	Before	After	Before	After
Particle size (nm)	144 ± 10.75	144.21 ± 8.31	144 ± 10.75	144. 45 ± 8.45
PDI	0.324 ± 0.03	0.326 ± 0.05	0.324 ± 0.03	0.326 ± 0.05
Zeta potential (mV)	−16.3 ± 0.07	16.21 ± 0.03	−16.3 ± 0.07	−16.15 ± 0.04
Entrapment Efficiency (%)	92.05 ± 6.12	92.01 ± 5.42	92.05 ± 6.12	92.02 ± 4.21
Loading Capacity	13.33 ± 0.71	13.22 ± 0.65	13.33 ± 0.71	13.11 ± 0.42

BQ: Bedaquiline: SLN: Solid lipid nanoparticles: SGF: simulated gastric fluid; SIF: simulated intestinal fluid; PDI: Polydispersity Index; nm: manometer.

**Table 11 pharmaceuticals-16-01309-t011:** In vitro gastrointestinal stability of BQ-loaded suspensions in various gastric media.

Stability Parameters	SGF (pH 1.2)		SIF (pH 6.8)	
	Before	After	Before	After
Particle size (nm)	1180 ± 22.25 nm	1180.21± 21.34	1180 ± 22.25 nm	1180.73 ± 20.23 nm
PDI	0.25 ± 0.04	0.27 ± 0.05	0.25 ± 0.04	0.29 ± 0.07
Zeta potential (mV)	−0.0668 ± 0.02	−0.0666 ± 0.07	−0.0668 ± 0.02	−0.0664 ± 0.02
Entrapment Efficiency (%)	88.89 ± 8.23	88.75 ± 7.45	88.89 ± 8.23	88.43 ± 5.21
Loading Capacity	11.43 ± 0.65	11.40 ± 0.41	11.43 ± 0.65	11.28 ± 0.67

SGF: Simulated gastric fluid; SIF: Simulated intestinal fluid; PDI: Polydispersity Index; nm: nanometer.

**Table 12 pharmaceuticals-16-01309-t012:** Pharmacokinetic parameters of the BQ suspension and BQ-SLN3.

Formulations	Pharmacokinetics Parameters					
BQ suspension	C_max_ (µg/mL)	T_max_ (µg/mL)	AUC (µg/mL/h)	MRT (h)	Ka (h^−1^)	T 0.5 (h^−1^)
	130 ± 3.12	1.0 ± 0.30	1128.5 ± 12.12	14.23 ± 0.35	5.04 ± 0.27	9.61 ± 0.25
BQ-SLN3	255 ± 6.13 *	1.0 ± 0.22 **	4412.4 ± 10.31 *	6.22 ± 1.01 *	2.8 ± 0.25 *	3.71 ± 0.31 *

BQ: Bedaquiline; SLN: Solid lipid nanoparticle: Data expressed as mean ± S.D. (n = 6); * Indicates *p <* 0.05 compared to the free drug group ** Indicates *p <* 0.01 compared to BQ suspension group.

**Table 13 pharmaceuticals-16-01309-t013:** Biochemical parameter evaluation in sera control, lung cancer group, and treated groups of rats.

Biochemical Parameters	Normal Control Group of Rats (I)	Lung Cancer Group of Rats (II)	Treated Group of Rats with BQ-Suspension (III)	Treated Group of Rats by BQ-SLN3 (IV)
MDA (nmol/mg)	50.55 ±3.87	80.21± 6.21 *	66.01 ± 3.51 **	54.2 ± 0.32 ***
SOD (unit/mg)	10.12 ± 0.81	4.21 ± 0.51 *	7.21 ± 0.31 **	9.5 ± 0.61 ***
CAT (nmol/min/mg)	905.12 ± 13.61	540 ± 9.12 *	752.1 ± 9.22 **	881.2 ± 6.21 ***
GSH (μg/mg)	4.51 ± 0.071	2.12 ± 0.12 *	3.51 ± 0.21 **	4.22± 0.32 ***
GR nmol/min/mg	5.21 ± 0.21	2.54 ± 0.22 *	3.82 ± 0.22 **	4.99 ±0.31 ***

n = 6; MDA: malondialdehyde; CAT: catalase: SOD: superoxide dismutase; GSH: glutathione; GR: glutathione reductase, * *p* < 0.05, ** *p*< 0.01 and *** *p* < 0.001 as limits of significance.

## Data Availability

The authors confirm that the data supporting the findings of this study are available within the articles and can be shared upon request.

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
