# Peer review of "Bedaquiline-Loaded Solid Lipid Nanoparticles Drug Delivery in the Management of Non-Small-Cell Lung Cancer (NSCLC)"

_pharmaceuticals, 2023, doi:10.3390/ph16091309_

Round 1

Reviewer 1 Report

The research has been framed in a very extensive manner. However the author may have a proof reading once to minimize several blunder errors.

1. 2.2.6 Stability studies (page 5) sample collection time interval is not matching with the results provided in 3.8 (Table 6).

2. Reference found to be missing for may procedure eg: 2.2.7 

3. 3.15 can be removed 

4. Many reference have not been formatted as per the author instruction.

5. in many cases minor errors such as italic (invitro, invivo) and subscription for chemical formula (H2O2)found to be error.

6. The order in methodology and its results found to be mismatching

7. No any clear context related to stability studies has been mentioned in the abstract part.

Need English correction with professionals, found many irrelevant phrases had been included to overcome plagiarism 

Author Response

Reviewer 1: Comments and Suggestions for Authors  

The research has been framed in a very extensive manner. However, the author may have a proof reading once to minimize several blunder errors.

  1. 2.6 Stability studies (page 5) sample collection time interval is not matching with the results provided in 3.8 (Table 6).

Author Response: Dear Reviewer, thank you for your remarks. It has been corrected now.

  1. Reference found to be missing for may procedure eg: 2.2.7 

Author Response: Dear Reviewer, thank you for remark. Now it has been corrected and checked thoroughly.

  1. 15 can be removed 

Author Response:  3.15. Experimental procedure, now it has been removed.  3.15 now is Pharmacodynamic evaluations in the manuscript.

  1. Many references have not been formatted as per the author instruction.

Author Response: Reference section has been revised as per the journal guidelines.

  1. in many cases minor errors such as italic (in vitro, in vivo) and subscription for chemical formula (H2O2) found to be error.

Author Response: Dear Reviewer, thank you for your remarks. Now fully checked in the manuscript and corrected now.

  1. The order in methodology and its results found to be mismatching

Author Response: Dear Reviewer, thank you for remarks. Now, it has been corrected.

  1. No any clear context related to stability studies has been mentioned in the abstract part.

Author Response:  Dear Reviewer, thank you for suggestion. Now, the stability part has been added in the abstract, discussion and conclusion section.

Comments on the Quality of English Language

Need English correction with professionals, found many irrelevant phrases had been included to overcome plagiarism 

Author Response: Dear Reviewer, thank you for remarks. We tried our best to fixed all such issues.

Reviewer 2 Report

This is an interesting study that explains using of Bedaquiline-loaded solid lipid nanoparticles for the treatment of non-small-cell lung cancer. In vitro characterization, cell viability, and cell culture and in vivo animal studies on bedaquiline-loaded solid lipid nanoparticles were also performed and the experimental methods of the study were clearly described. However, the manuscript has suffered some major issues. Some of the points require further attention and the content is eligible to publish after the authors would manage with the comments given below.

- An oral suspension of Bedaquiline was also developed for comparison in the study. However, no characterization studies have been performed on it. On the other hand, the author should explain how they would like to apply SLN by oral route. What is the final product?

- The authors should explain why they use 37oC for solubility studies. (Page 3; 2.2.2.1. Solubility study)

- Is it enough to add fresh dissolution medium to maintain sink condition? (Page 5, line 205).

 - In the in vitro release studies, the pH of the release medium is 7.2. Dissolution studies should be carried out at different pHs to mimic the gastrointestinal tract.

- According to 2.2.6. Stability studies, to evaluate the stability of the BQ-loaded SLN, samples were stored for a total of twelve weeks at 4°C, 25°C, and 37°C. However, Long-term stability test findings for BQ-loaded SLN at 25°C/60% RH were presented. (Page 12, Line 398).

- The authors said that “The optimized BQ-SLN formulation, which had a particle size of 144 nm and PDI of 0.324, is shown in Figure 2A. This demonstrated the monodisperse nanostructure of the BQ-loaded SLN formulation.” (page 9, line 366) However, according to Figure 2A, the particle distribution has two peaks which means that there are many aggregates. The authors should explain this.

Some minor editing of English language required

Author Response

Reviewer 2: Comments and Suggestions for Authors

This is an interesting study that explains using of Bedaquiline-loaded solid lipid nanoparticles for the treatment of non-small-cell lung cancer. In vitro characterization, cell viability, and cell culture and in vivo animal studies on bedaquiline-loaded solid lipid nanoparticles were also performed and the experimental methods of the study were clearly described. However, the manuscript has suffered some major issues. Some of the points require further attention and the content is eligible to publish after the authors would manage with the comments given below.

Author Response: Thank you very much for your remark. We tried our best to address all the given points

 - An oral suspension of Bedaquiline was also developed for comparison in the study. However, no characterization studies have been performed on it. On the other hand, the author should explain how they would like to apply SLN by oral route. What is the final product?

Author Response: Dear Reviewer, Thank you for worthy remarks. The characterization of Bedaquiline suspension has been incorporated now.

- The authors should explain why they use 37oC for solubility studies. (Page 3; 2.2.2.1. Solubility study)

Author Response: Dear reviewer, thank you for the remark. By mistake 37oC has been given. It has been corrected. Correct temperature is 25 ± 1°C.

- Is it enough to add fresh dissolution medium to maintain sink condition? (Page 5, line 205).

Author Response: Corrections have been made in the manuscript.

 - In the in vitro release studies, the pH of the release medium is 7.2. Dissolution studies should be carried out at different pHs to mimic the gastrointestinal tract.

Author Response: Dear Reviewer, thank you for your remarks. I agreed, now provided in the manuscript.

- According to 2.2.6. Stability studies, to evaluate the stability of the BQ-loaded SLN, samples were stored for a total of twelve weeks at 4°C 25°C, and 37°C. However, Long-term stability test findings for BQ-loaded SLN at 25°C/60% RH were presented. (Page 12, Line 398).

 Author Response: Dear Reviewer, thank you for remarks. Due to our mistake this temperature is written. Now corrections have been made, the stability studies evaluated at 25°C/60% RH and 40°C/75% RH for a total of 90 days (twelve weeks). Long term studies conducted at 25°C/60% RH, whereas the stressed stability tested at 40°C/75% RH.

- The authors said that “The optimized BQ-SLN formulation, which had a particle size of 144 nm and PDI of 0.324, is shown in Figure 2A. This demonstrated the monodisperse nanostructure of the BQ-loaded SLN formulation.” (page 9, line 366) However, according to Figure 2A, the particle distribution has two peaks which means that there are many aggregates. The authors should explain this.

Author Response: I totally agreed on your raised points. Corrected Fig 2A has been provided now.

Reviewer 3 Report

the manuscript entitled "Bedaquiline-loaded solid lipid nanoparticles drug delivery in 2 the management of non-small-cell lung cancer (NSCLC)" by Ullah et al is remarkably interesting and novel. The author has conducted extensive studies and the results are presenting good. I strongly recommend this manuscript for publication in the current form.

Author Response

Reviewer 3: Comments and Suggestions for Authors

the manuscript entitled "Bedaquiline-loaded solid lipid nanoparticles drug delivery in 2 the management of non-small-cell lung cancer (NSCLC)" by Ullah et al is remarkably interesting and novel. The author has conducted extensive studies and the results are presenting good. I strongly recommend this manuscript for publication in the current form.

Author Response: Dear Reviewer, Thank you.

Round 2

Reviewer 2 Report

It is suitable for publication.

Minor editing of English language required